# Giant Bilateral Iliopsoas Abscesses, Secondary to Pott’s Disease: Challenging Diagnostic–Therapeutic Protocol Management (Modern and Innovative Open Approach Technique Through J.L. Petit Triangle)—A Case Report and Literature Review

**DOI:** 10.3390/diagnostics15192506

**Published:** 2025-10-02

**Authors:** Mihaela D. Pîrvu, Cristian C. Popa, Iulian Lupu, Cătălin N. Grasa, Anca T. Gheorghe, Vasile Sârbu

**Affiliations:** 1First Surgery Department, Emergency County Clinical Hospital, 900591 Constanța, Romania; 2Department of General Surgery, Faculty of Medicine, “Ovidius” University, 900470 Constanța, Romania; 3Department of General Surgery, “Carol Davila” University of Medicine and Pharmacy, 050474 Bucharest, Romania; 4Second Department of Surgery, University Emergency Hospital Bucharest, 050098 Bucharest, Romania; 5Osteoarticular TB Clinic, Emergency County Clinical Hospital, 900591 Constanța, Romania; 6Medical Sciences, The Academy of Romanian Scientists, 010702 Bucharest, Romania

**Keywords:** iliopsoas, abscess drainage, Petit’s triangle, tuberculosis, Pott’s disease, lumbago sciatica

## Abstract

**Background and Clinical Significance:** Iliopsoas abscess has recently become a condition quite frequently present in our practice, arising through hematogenous or lymphatic dissemination (primary), or secondary to trauma or infectious–inflammatory vertebral, renal, or gastrointestinal diseases. It is often diagnosed with difficulty, due to the insidious and rather atypical symptomatology. The simultaneous relevance to neurosurgery, orthopedics, urology, rheumatology, and of course surgery, makes the iliopsoas abscess a real challenge in diagnosis and treatment for any of us, as well as collaboration between specialties. **Case Presentation**: The aim of this paper is to illustrate all this through a rare clinical case of vertebral tuberculosis, with giant abscesses of bilateral iliopsoas and comparison with data from the literature, through a review. **Conclusions**: The problems were related to the clinical–paraclinical, etiological diagnosis, surgical strategy, technique and tactics, surgical approach, treatment and immediate, and both long-term postoperative management. The ultimate goal is to reduce morbidity and mortality secondary to this often-disabling condition.

## 1. Introduction

Iliopsoas muscle abscess is a serious but fortunately quite rare condition (with a reported incidence of 0.4/100,000 in the UK [1]), with a higher prevalence in males than females. The average age is between 44 and 58 years, depending on the geographical region. Regarding the location of the condition, cases found on the right side are relatively equal to the left, with bilateral abscesses appearing exceptionally [2]. Iliopsoas muscle abscess is due to systemic or neighboring infections, in the context of predisposing factors, such as immunosuppression (e.g., HIV, neoplasms, drug use) [1]. Since it is a retroperitoneal infection, with vague and non-specific symptoms (fever, back or lower limb pain, static, and kinetic disorders) [3,4], it is often difficult and late in establishing an etiological, differential, and definitive diagnosis. However, due to the evolution of the performance of current imaging methods (CT, MRI), the possibility of identifying this pathology has increased. As a result, cases of iliopsoas abscess have become increasingly present in current medical practice in the literature [5].

Treatment consists of drainage of the abscess, antibiotic therapy, treatment of etiopathogenic condition, supportive treatment, management of atypical pain and complications that may occur (e.g., sepsis). Drainage of the abscess can be performed in several ways, depending on etiology, pathogenesis, location, size, and severity. It can be minimally invasive—percutaneous imaging guided, laparoscopic, and open surgical.

We aim to point out the difficulties of diagnosing cases of iliopsoas abscesses secondary to Pott’s disease and to promote open surgical drainage via the J.L. Petit lumbar triangle. We preferred the lumbar approach, through J.L Petit triangle, because it offers direct drainage of the abscess with proven efficacy, minimizing the risk of hemorrhagic or infectious complications and recurrences.

The particularities of this case consist of the fact that the abscesses were gigantic and bilateral, which we have not encountered in the specialized literature, and the novelty of drainage through the Petit triangle, an innovative and modern technique, which has also not been cited.

The ultimate goal is to reduce morbidity and mortality secondary to this often-disabling condition.

## 2. Case Report

Regarding our personal experience, we present the case of a young patient, 32 years old, with grade III obesity, who was admitted to our clinic in February 2023. We used the data extracted from observation sheets collected between February 2023 and December 2024.

The patient, who had been in contact with a colleague with tuberculosis in 2009, underwent preventive tuberculosis treatment.

In 2016, she was diagnosed with a giant left frontal, multilocular brain abscess, clinically manifested by right hemiparesis and moderate meningeal syndrome. A left frontal flap and evacuation were performed. The evolution was favorable, with a return to normal consciousness and the disappearance of the motor deficit. The Quantiferon-TB gold test was positive, but the patient did not undergo anti-tuberculosis treatment.

In 2019, she had epileptic seizures, and a CT scan revealed a right frontal lobe sequela. Chronic treatment with carbamazepine was recommended.

In 2021, the episode of illness started, with progressive evolution, with the impairment of walking, which became possible only with support, unilateral, antalgic walking.

In February 2023, the patient was admitted to the clinic. An MRI of the lumbar spine was performed upon admission. It revealed multiple injuries, but with the preservation of certain structures intact: erosive changes, with association of fluid densities, located at the level of the intervertebral disks (diskitis L4–L5 and L5–S1); multiloculated collections in the psoas muscles bilaterally, 230/79 mm (cc/ll) left and 200/82 mm (cc/ll) right, without clear signs of extension at the epidural space; vertebral bodies, aligned at the anterior and posterior wall; overlying intervertebral disks with preserved height and not exceeding the vertebral contour (Figure 1).

Upon admission, from clinical point of view, the patient was afebrile, but with bilateral lumbar and lower limb pain, with functional impotence of the left lower limb and algoparesthesia of the left thigh in the projection territory of L3–L4. The patient was unable to mobilize or move the left lower limb or maintain an antalgic semi-sitting position in bed, with the left lower limb semiflexed and the hip externally rotated.

Laboratory tests were performed in an emergency setting revealed a moderate inflammatory syndrome (fibrinogen—675 mg/dL, C-reactive protein 11.7 mg/dL), but with a normal leukocyte count (8.37 × 10^3^/µL µL), with discrete neutrophilia (77.2%) and a moderate anemic syndrome with hemoglobin 9.1 g/dL and hematocrit 28.7% (Table 1). Testing for anti-HIV1+2 was negative. The tests for Hepatitis B surface antigen (AgHBs) and anti-HCV antibodies were also negative (Table 1). The Quantiferon-TB gold test was also negative (Table 2). This could be secondary to immunosuppression, laboratory technique (incorrect collection, transport or processing of the blood sample—delay or insufficient shaking of the containers), problems with the reagents used or calibration of the equipment, extrapulmonary tuberculosis (the immune response may be altered, in this situation, leading to a false negative result), human or administrative error (misinterpretation of results or confusion of evidence).

Based on clinical data, laboratory test results, and imaging investigations, a diagnosis of giant bilateral iliopsoas abscesses was established, and emergency surgery was undertaken.

The surgical intervention, performed under general anesthesia with orotracheal intubation and mechanical ventilation, consisted of making an incision of approximately 4 cm, above the iliac crest, in the middle 1/3, at the level of the J.L. Petit lumbar triangle (Figure 2). The incision was made behind the external oblique and anterior to the transverse abdominis. The thoracolumbar fascia (origin of the internal oblique and the transverse abdominis) was digitally punctured. It was penetrated between the external oblique and the latissimus dorsi muscle. Then, it was penetrated between the iliac crest and the quadratus lumborum, under the anterior face of the iliac bone, then with a finger the psoas sheath was punctured at the lateral edge (Figure 3). Thus, the abscess cavity was drained. A large amount of purulent fluid was evacuated—500 mL on the left side and 300 mL on the right side. Then excisional debridement and abdominal compression for complete drainage were performed. The two cavities were drained, each with two drain tubes, one 28 Fr silicone and a 22 Fr Foley catheter, through which abundant lavage was performed with bactericidal, antiseptic substances (Figure 4). A compression dressing was then applied.

The postoperative evolution was rapidly favorable. The general condition improved rapidly, the purulent drainage decreased, later becoming seropurulent and serous-fibrinous, in just a few days. The lumbar pain and pain in the bilateral lower limb improved, as well as the algoparesthesia and functional impotence of the left lower limb. The antalgic, semi-sitting position in bed, with the left lower limb in semiflexion and the hip externally rotated, disappeared immediately postoperatively. Mobilization and movements of the left lower limb quickly became possible, initially with a walking frame. However, the neurosurgeon’s recommendations were for absolute bed rest, followed by lumbar orthosis until the discitis lesions healed—approximately 6 months.

The results of the bacteriological examination revealed the following: pus culture—no bacterial flora developed on the culture media, Koch’s bacillus (BK) microscopic examination—negative, ADA (adenosine deaminase) test in pus—negative (Table 3).

Two days postoperatively, a contrast-enhanced MRI revealed spondylodiscitis L4, L5, with L5 trans-somatic extension; small prevertebral collection; anterior epidural infection L4–S1; resolution of accumulations in the psoas major, regressive iliac abscess; and left coxofemoral osteoarthritis, characterized by pinching of the left coxofemoral joint space, with subchondral edema (Figure 5).

Neurosurgical examination noted the absence of signs of surgical root compression and recommended conservative treatment.

Drainage of the abscesses was maintained for three weeks, with sequential tube removal. During this time, lavage with betadine serum and rifampicin was performed.

The patient was transferred to the orthopedics department, osteoarticular tuberculosis unit, for further treatment.

Conservative treatment was centered on initiating empirical anti-tuberculosis treatment (regimen I), represented by the standard first-line treatment regimen. This must be administered daily, for seven days a week, extended up to 12 months (compared to 6 months, in the case of pulmonary tuberculosis), due to the weaker penetration of drugs into bone tissue. Treatment was administered in two phases: initially an intensive two-month phase, with daily isoniazid, rifampicin, pyrazinamide and ethambutol (in order to reduce the bacterial load), then a continuation phase, with hydrazide and rifampicin (elimination of latent bacilli to prevent relapse), according to World Health Organization guidelines [6].

The patient was discharged one month after admission, with recommendations for absolute bed rest, wearing a lumbar brace, and clinical and paraclinical re-evaluation after 6 weeks.

After discharge, three months later, the result of the culture on Lowenstein–Jenson medium for Koch’s bacillus was obtained, which came out positive for the *Mycobacterium tuberculosis* complex (AGMPT64—positive), and the antibiogram revealed that the identified bacillus species is sensitive to hydrazide and rifampicin.

The patient was re-admitted and re-evaluated periodically: initially after six weeks (April 2023), then after another three months (July 2023), respectively, after six months (February 2024), then after another seven months (September 2024) for clinical–biological evaluation. The effectiveness of the treatment was confirmed by the biological samples, which were within normal limits. From an imaging point of view, the lumbar radiography revealed a stabilization process of the infectious process through the fusion of L4–L5. Gradually, she was recommended muscle toning exercises without overloading the spine, monitoring the weight curve, medical gymnastics, then physiotherapy for general toning. She was also recommended psychological and psychiatric supportive treatment.

Currently, the patient limps, has left hip pain, and analgesic limitation of mobility.

A new MRI, performed in October 2024, revealed L4–L5 spondylodiscitis with intradiskal and trans-somatic L5 collection. A unilocular collection with fluid–parafluid content was also observed, restrictive on diffusion-weighted imaging, with its own wall with irregular thickness of 3–5 mm, measuring approximately 28/15 mm in the axial plane. This developed along the iliopsoas, from the place of formation to the left coxofemoral joint, where the latter level associating gadolinium contrast synovial reaction, erosions, and edema of the femoral cephalic region were noted, extending to the trochanteric region.

A final reassessment, carried out in December 2024, highlighted a slowly favorable evolution, with recommendations continuing supportive treatment, gentle joint exercises, and anti-arthritis therapy.

## 3. Discussion with Literature Review

For a good understanding of the described pathology, we have highlighted several essential aspects, comparing our results with data from the literature.

### 3.1. Etiopathology Issues

Regarding etiopathogenesis, iliopsoas abscesses can be primary or secondary. They are considered primary when they occur through systemic, hematogenous, or lymphatic transmission of infection. Secondary abscesses develop from loco-regional, intra-, or retroperitoneal infections, from the organs with which it is related, including gastrointestinal diseases (diverticulitis, Crohn’s disease, retrocecal appendicitis), urogenital diseases (pyelonephritis, secondary urinary tract infections, ureteral lithiasis, tubo-ovarian abscesses), or bone diseases (rheumatoid disease, septic arthritis, osteoarticular tuberculosis, polyserositis, gout, etc.), tumors or trauma, by their extension [2]. In patients without apparent immunosuppressive conditions, secondary iliopsoas abscesses are exceptionally rare. A 2021 PubMed database search identified only nine cases reported in the literature [3,5,7,8,9,10,11,12,13,14]. Because this pathology is extremely rare, it has only been cited in the literature as case reports.

Tuberculosis is a common infectious disease caused by various types of mycobacteria, most commonly, *Mycobacterium tuberculosis*. It usually affects the respiratory system, in 79% to 87% of cases, but can also affect other parts of the body. At the osteoarticular level, it occurs in 1–3% of tuberculosis cases, with the spine being the most commonly affected (approximately 50% of cases). Other locations include the pelvis, knees, and peripheral joints [15,16]. It can cause lesions in the bones, ligaments and joints, lesions that can extend loco-regionally and affect various other structures. One of these lesions is represented by the abscess of the iliopsoas muscle.

Tuberculosis is the most common pathology involved in decreased immunity and the development of iliopsoas abscesses. Although extremely rare, but not negligible, iliopsoas abscess can occur secondary to extra-articular tuberculosis. The literature cites an incidence of 3–5% of all cases of tuberculosis. It is also worth noting the incidence of primary muscle tuberculosis, which is very rare, 0.015% of cases [14]. Iliopsoas abscess occurs most frequently from an active cold paravertebral abscess.

Of all, inflammatory conditions of the spine, which include vertebral osteomyelitis or epidural abscess, are increasingly incriminated in etiology, due to the modern lifestyle, with increased mobility, especially among active patients [17].

### 3.2. Anatomical Particularities

From an anatomical point of view, the iliopsoas muscle is formed by the union of the psoas major muscle (which is later joined by the psoas minor muscle) with the iliac muscle. The psoas major muscle originates on T12, L1, L2, L3, and L4 and, in this way, a single muscular body is formed, which crosses the lumbar and pelvic region, exits the pelvis through the muscular gap under the inguinal ligament, and reaches the thigh. The iliac muscle inserts upwards on the iliac fossa and passes together with the psoas major through the muscular gap. The two united muscles insert via a strong common tendon on the lesser trochanter of the femur (Figure 6).

Another important local–regional anatomical element is the iliopectineal synovial bursa. It is inserted on the anterior edge of the coxal, respectively, on the deep surface of the iliopsoas muscle and has a role of protection and facilitation of appropriate local movements. In 10% of cases, it communicates with the coxofemoral joint cavity, which explains the infections of the joint transmitted from purulent collections that propagate along the iliopsoas muscle.

The iliopsoas muscle has numerous relationships with surrounding structures. Thus, in the abdominopelvic region, it is anteriorly related to the psoas minor muscle, with which it unites, but also to various retroperitoneal abdominal organs: the kidney, renal vessels, ureter, spermatic/ovarian vessels, ascending or descending colon; posteriorly, it is related to the quadratus lumborum and the costiform processes of the lumbar vertebrae; on the medial edge, it is related to the external iliac vessels, alongside which it descends into the pelvis. The iliac muscle has an anterior relationship with the cecum on the right or the iliopelvic colon on the left [18].

The lower lumbar triangle, described by Jean Luis Petit in 1973, is located between the iliac crest, the external oblique muscle, and the latissimus dorsi muscle and is covered by the superficial lumbar fascia. It is a scientifically important area for several reasons. In anatomy and surgery, it represents an area of weakness in the posterior abdominal wall, an area susceptible to hernias (Petit hernias), one of the three type of lumbar hernias described: through the upper lumbar triangle (Grynfeltt–Lesshaft lumbar hernia), through the lower lumbar triangle (Petit and diffuse), in which it is not possible to clearly delineate between the two) (Figure 7 and Figure 8). In imaging, especially in computed tomography, the anatomical understanding of this region helps to differentiate hernias from other lumbar pathologies, such as abscesses and tumors. The study of this region has contributed to minimally invasive approaches, including image-guided, laparoscopic, and in reconstructive surgery [19,20].

### 3.3. Diagnoses Management

The main risk factors are exposure to patients infected with tuberculosis, poor socio-economic background, certain immunodeficiencies (diabetes, HIV, viral hepatitis, autoimmune diseases), alcohol or drug abuse, and the use of immunosuppressive drugs.

The main microbial agents involved in the etiology of iliopsoas abscess are represented by *Staphylococcus aureus*, *Streptococcus* and *Escherichia coli*. *Mycobacterium tuberculosis* is a commonly found bacterium in psoas abscess in geographic regions with a high prevalence of tuberculosis. Other etiological factors include *Mycobacterium kansasii*, Langerhans cell histiocytosis (in children and adolescents), paracoccidioidomycosis, nocardiosis, and other granulomatous bacteria (brucellosis) [21,22].

The symptoms may be different. A classic presentation, described in clinical studies, includes fever, low back pain, and lameness, but these occur in only 30% of cases. Weight loss, altered general condition, weakness, pain in the lower limbs, and functional impotence may also occur [2].

General and local clinical examinations can reveal algoparesthesia, the impossibility of mobilization and movement of the lower limbs, with an antalgic position.

The laboratory examination is based on laboratory and imaging data. Laboratory data may reveal an inflammatory syndrome with or without leukocytosis, with increased fibrinogen, C-reactive protein, erythrocyte sedimentation rate, and a variable degree of anemia. The Quantiferon-TB gold test may detect tuberculosis. It is a modern diagnostic method, which uses IGRA technology (Interferon Gamma Release Assay). The test quantifies the cell-mediated immune response to tuberculosis infection by measuring the amount of interferon gamma released following stimulation with two mixtures of antigens, which contain peptides similar to the specific proteins ESAT-6 and CFP-10 (AgTB1 and AgTB2). AgTB1 aims to induce an immune response by CD4+ helper T lymphocytes, while AgTB2 aims to induce an immune response by CD8+ cytotoxic T lymphocytes, the latter being more frequently associated with active tuberculosis infection and recent exposure to the pathogen. When interpreting the results, the values obtained for interferon gamma in the presence of the negative control (Nil) and the positive control (mitogen) are taken into account. The result in our patient does not indicate a secretion of interferon gamma towards the tuberculosis-specific antigen.

The Quantiferon-TB gold test is useful in screening for latent tuberculosis. It has a sensitivity of 80–90%, specificity over 95–99%. Like any test, it has its limitations. The test does not differentiate between latent or active infection. False negative results may occur in cases of diseases that affect immunity or secondary to errors in laboratory technique (delayed or incorrect collection, transport or processing of the blood sample, insufficient shaking of the containers), problems with the reagents used or equipment calibration. Our situation could also be due to extrapulmonary tuberculosis, in which the immune response was altered, which could lead to a false negative result. The negative test could also be due to human or administrative error (misinterpretation of results or sample mix-up) [23,24,25,26,27].

The gold standard in imaging diagnosis is MRI (pelvic), but CT is also a valuable imaging investigation, especially in emergencies, which, in addition, provides important information about the state of intra- or extraperitoneal organs, helping us to establish an etiological diagnosis. Both, performed with contrast medium, can provide additional information related to the loco-regional vascularization, spine, intervertebral disks, spinal cord, bony pelvis, pelvic, and intra-abdominal organs, etc.

The pathophysiological diagnosis can most likely be explained by secondary hematogenous spread, through the paraspinal vessels, from a primary focus of infection located in the lungs or extrapulmonary.

Destruction of the vertebrae leads to kyphotic deformities of the spine, with instability, debilitating pain, muscle weakness of severe neurological origin, progressive neurological deficit, leading to paresis and paralysis [21].

### 3.4. Therapeutic Management

Therapeutic management is sequential and performed in several stages.

Abscess drainage can be performed surgically or minimally invasive, guided percutaneously by imaging (ultrasound, CT) or endoscopically (laparoscopic) [28,29,30].

Image-guided drainage can also be performed under local anesthesia or sedation and has the following advantages: lower anesthetic–surgical risk than open surgery. It would be indicated in selected cases, in patients with immediate major vital risk (e.g., sepsis), until a curative intervention can be performed: patients with multiple comorbidities, in serious condition, with maximum anesthetic–surgical risk or in selected cases—well-defined, unilocular, well-defined abscess, located in an accessible position, with fluid collection and microbiological investigation of it (cultures with antibiogram). Prior imaging investigation is required, such as ultrasound or CT with precise determination of the location of the abscess, dimensions, appearance. Abscesses can be drained by needle or catheter puncture. It can be left in place and washing–drainage can be performed if necessary. There advantages include low costs, lower risks, and simplicity. However, if the abscess is old, has dense multiloculate content, is deep, or in the vicinity of vital structures (vessels, nerves), anatomical access may be difficult and limited, and this type of drainage may be inadequate and insufficient [29,30].

Classical drainage is recommended when the previous ones fail, or are not possible, in large, multilocular abscesses, associated with necrosis. It is also indicated in extensive lesions, severe complications such as peritonitis, sepsis, vertebral bone tuberculosis, and retroperitoneal tumors [31,32].

Laparoscopic drainage is performed under general anesthesia by orotracheal intubation and mechanical ventilation—it is less used than image-guided drainage in selected cases, large multilocular abscesses and which, more frequently, associate other intra-abdominal pathologies: appendicitis, diverticulitis, pelvic inflammatory disease, and peritoneal tuberculosis. Laparoscopic drainage is indicated in more complex cases, particularly when an intraperitoneal pathology is present or when image-guided drainage has not provided results [33,34,35].

In turn, abdominal drainage can be open (transperitoneal or extraperitoneal) [32]. Open surgical drainage can be carried out via a retroperitoneal approach (through the iliac fossa), or lumbar approach in cases of abscesses of renal or intraperitoneal etiology, often performed simultaneously with treatment of a primary gastrointestinal cause (e.g., appendicitis, diverticulitis) [21].

The surgical technique preferred and promoted by us is based on lumbar surgical drainage, through the J.L. Petit triangle. This approach provides direct access to the iliopsoas muscle and allows for effective drainage of the abscess, reducing the risk of complications and recurrence. Drainage through Petit’s triangle can be maintained for a longer period of time without the risk of major bleeding or retroperitoneal superinfection. Lavage and drainage is easy, with antiseptics or anti-tuberculosis drugs (e.g., rifampicin). Recurrent abscess or relapses (less frequent than in image-guided drainage) can be resolved more easily by re-catheterization of the superficial orifice with a drain tube.

We preferred to drain the remaining abscess cavity through the postoperative wound for several reasons as follows:✓The rationale for placing trans-incisional drain tubes was to maintain as wide a direct communication as possible with the remaining abscess cavity for as long as possible and to promote healing from deep to superficial.The drainage of the abscess should be as wide and direct as possible to maintain the widest, lowest, and longest access possible for continuous, easy, and efficient postoperative lavage and drainage.The advantage of transforming into a tunnel over time, with progressive scarring, which, if necessary, could even allow easier re-catheterization.The anatomical peculiarity of the region, small in size, which can be approached safely without major risks, practically corresponding to the incision made.An additional incision in a septic area increases the risk of further contamination of that region.

This anatomical corridor is an efficient and safe option for bilateral drainage in spinal tuberculosis, as evidenced by the good evolution, with the cure of the patient presented.

It is mandatory to collect fluid, from which to perform culture with antibiogram, cytology, microscopic analysis for Koch’s bacillus, adenosine deaminase test, and biochemical analysis. We also consider performing culture for Koch’s bacillus, on Lowenstein—Jenson medium with antibiogram, mandatory in any suspicion of tuberculosis, bone, or joint damage.

Antimicrobial treatment should be guided by the results of the culture and antibiogram. Until the results are obtained or when they are negative, anti-staphylococcal antibiotic therapy (in the case of primary abscess) or broad-spectrum antibiotics (in the secondary one) are preferred.

Anti-tuberculosis chemotherapy should be started empirically when there is high suspicion. If cultures confirm Koch’s bacillus, then therapy is adjusted accordingly [36].

Therapeutic management may also include spinal interventions, performed by neurosurgeons, through anterior, posterior, or combined approaches to drain collections, debride necrotic and infected tissues, stabilize the vertebrae, correct deformities, or fusion with implants [37,38].

Physical and functional rehabilitation is essential and should be performed in stages. Initially, a lumbar orthosis; then, muscle toning exercises without overloading the spine; followed by mobility exercises, medical gymnastics, and physiotherapy for general toning.

Psychological and psychiatric supportive treatment is very important, because recovery is long and difficult and only in this way can patients cope with stress and anxiety [39].

Last but not least, proper management of Pott’s disease requires patient compliance [40].

It is important that iliopsoas muscle abscess is correctly identified and properly treated, otherwise it can become a mutilating condition, with important socio-economic consequences.

## 4. Conclusions

Due to the high incidence of lumbosciatica in current medical practice, it is recommended that any suspicion regarding this pathology is investigated comprehensively, both clinically and especially via imaging—MRI, CT.

We recommend the surgical approach through the Petit lumbar triangle for the drainage of iliopsoas abscesses, because we believe that it offers important advantages that reduce morbidity and mortality.

The existence of a complex interprofessional team, consisting of a surgeon, neurosurgeon, infectious disease specialist, pulmonologist, orthopedist, urologist, rheumatologist, psychologist, physiotherapist, and their effective collaboration in the management of Pott’s disease treatment, is very important.

A well-developed follow-up plan is also necessary, which includes the tuberculosis specialist, but also other specialties, including primary care. If not diagnosed in time and not treated properly, bone tuberculosis, in general, and secondary iliopsoas abscess in particular can become a mutilating condition, with important socio-economic consequences.

Bone tuberculosis remains an important public health problem, which involves a great need for diagnostic and therapeutic resources, with important consequences on patient morbidity and mortality.

## Figures and Tables

**Figure 1 diagnostics-15-02506-f001:**
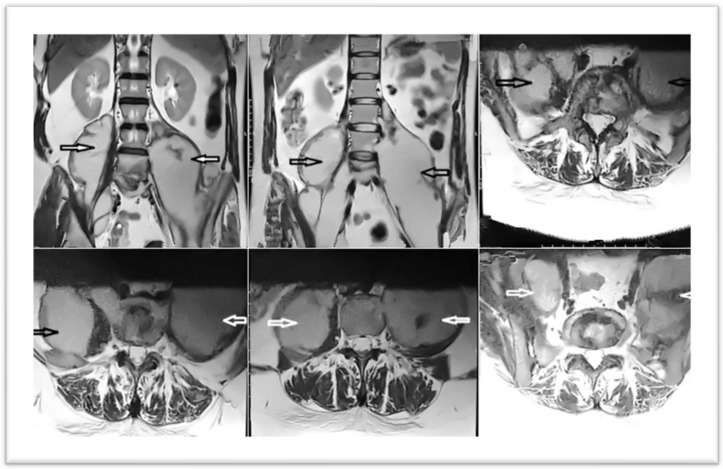
MRI images at admission—sagittal and transverse sections—arrows indicate the two collections, left and right.

**Figure 2 diagnostics-15-02506-f002:**
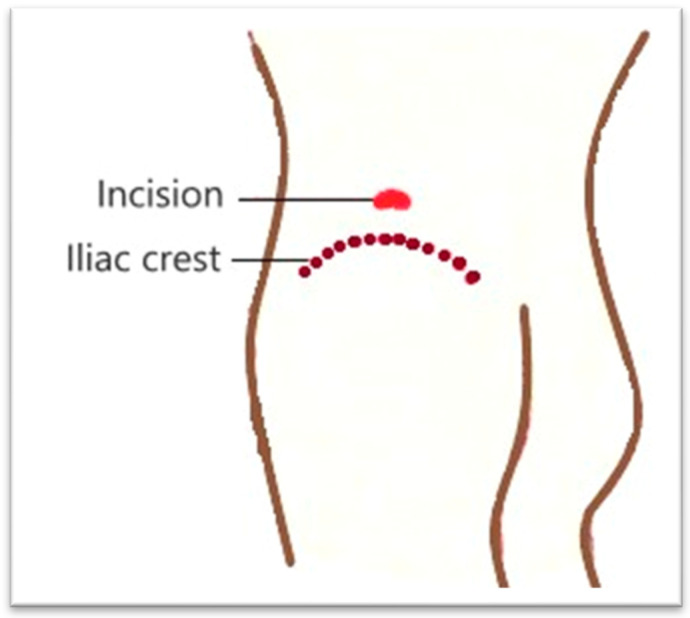
Incision site.

**Figure 3 diagnostics-15-02506-f003:**
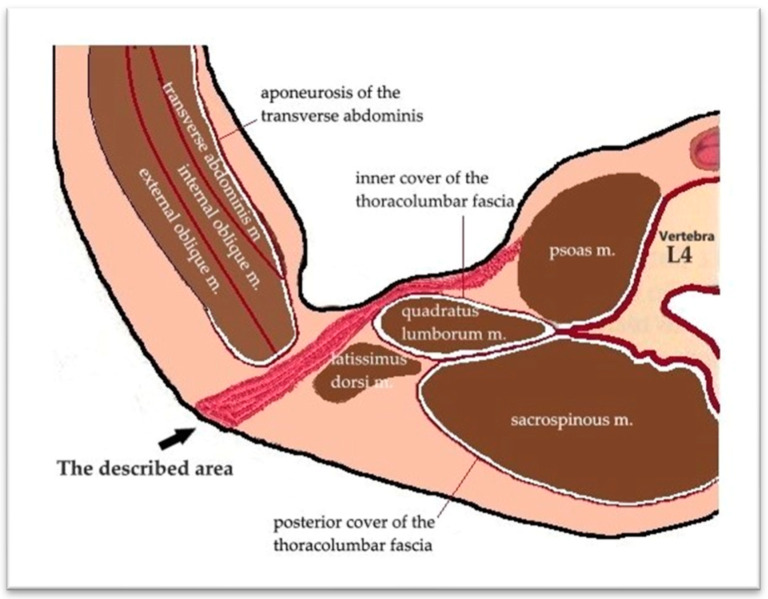
The surgical approach.

**Figure 4 diagnostics-15-02506-f004:**
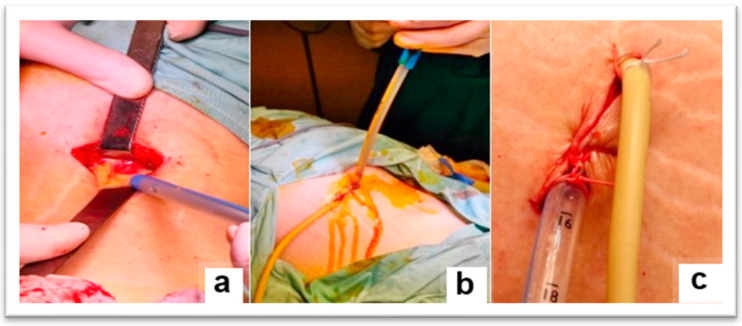
Intraoperative images—(**a**) incision above the iliac crest, in the middle 1/3, at the level of the J.L. Petit lumbar triangle, (**b**) double drainage of the remaining cavity, (**c**) final aspect, after fixation of the drainage tubes (silicone tube and Foley catheter).

**Figure 5 diagnostics-15-02506-f005:**
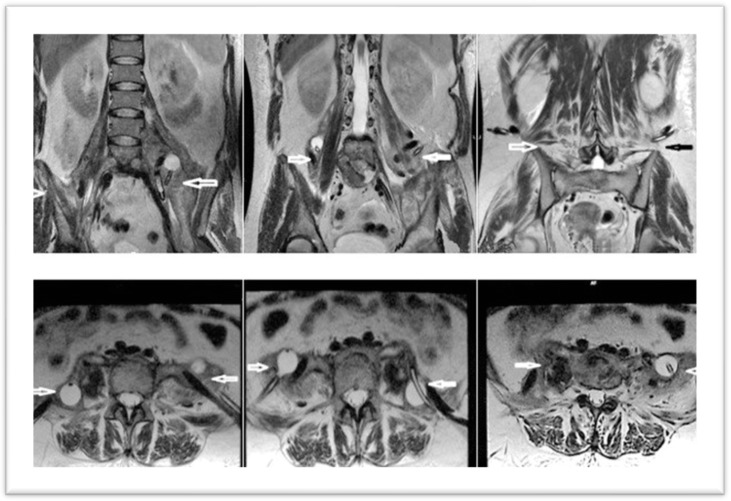
MRI aspect—two days postoperatively (arrows indicate resolution of abscesses under drainage).

**Figure 6 diagnostics-15-02506-f006:**
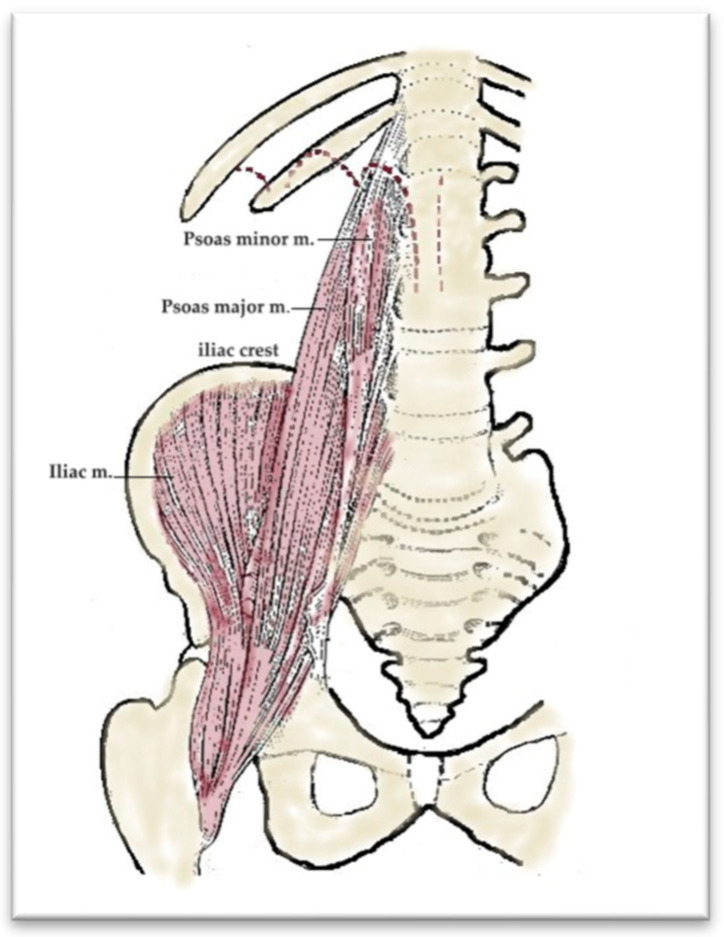
Anterior view—Iliopsoas muscle.

**Figure 7 diagnostics-15-02506-f007:**
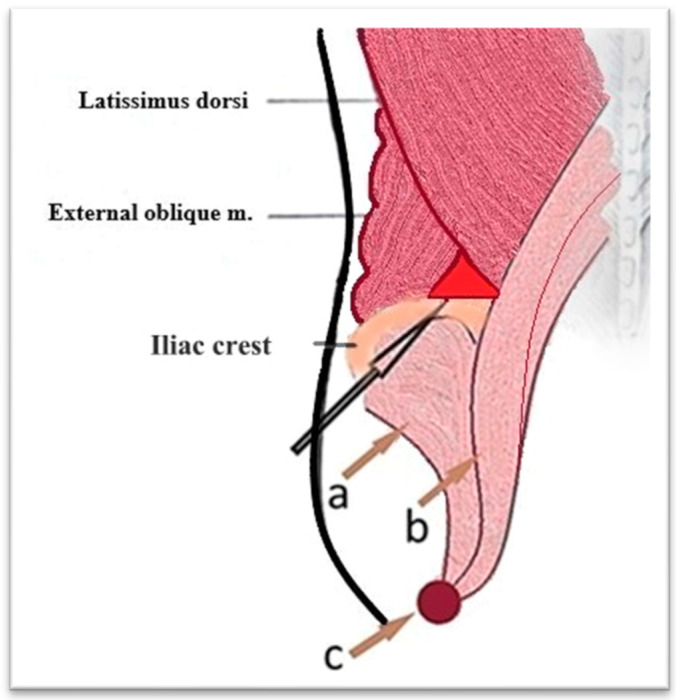
Inferior Petit lumbar triangle—the black arrow indicates the area of the proposed approach; brown arrows indicate the projection of iliacus muscle (a), the greater psoas muscle (b), and the common insertion on the lesser trochanter of the femur (c).

**Figure 8 diagnostics-15-02506-f008:**
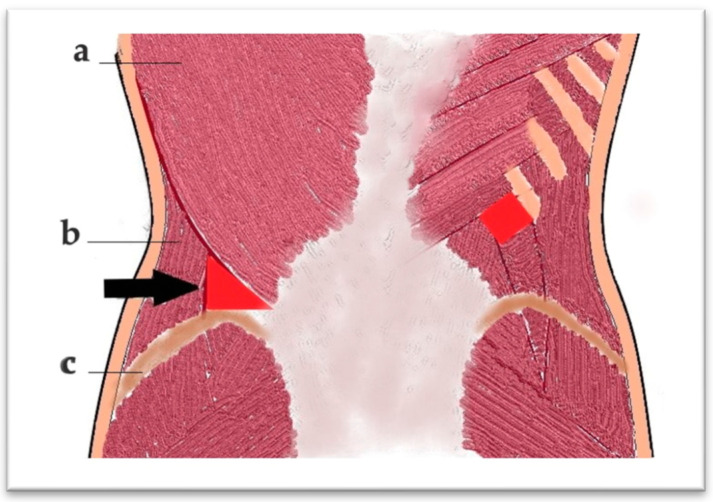
Posterior view—arrow indicates the Petit lumbar triangle. Latissimus dorsi m. (a), external oblique m. (b), iliac crest (c).

**Table 1 diagnostics-15-02506-t001:** Laboratory analyses.

Analysis	Value	Reference Range	–UM-
No. leukocyte	8.37	3.6–10^3^	µL
Neutrophil	77.2	34–71	%
Hemoglobin	9.1	12–18	g/dL
Hematocrit	28.7	36–54	%
C-reactive protein	11.7	<0.4	mg/dL
Fibrinogen	675	200–400	mg/dL
HBsAg-	negative		
AgHBs	<2	<10 (absence of immunity)	UI/L
Anti-HIV1+2	negative		

**Table 2 diagnostics-15-02506-t002:** Laboratory analyzes—The Quantiferon-TB gold test.

Analysis	Value	Reference Range	–UM-
Amount of interferon gamma released (1)	0.328	<0.35	UI/mL
Amount of interferon gamma released (2)	0.322	<0.35	UI/mL
Mitogenic control	8.402	≥0.5	UI/mL
Negative control (Nil)	0.053	≤8	UI/mL
Test result	negative	negative	

**Table 3 diagnostics-15-02506-t003:** Laboratory analyzes—examination of the collected fluid.

Analysis	Result
Culture	Negative
BK microscopic examination	Negative
ADA (adenosine deaminase) test	Negative
Radiometric BACTEC cultures	Negative

## Data Availability

The original contributions presented in this study are included in the article. Further inquiries can be directed to the corresponding authors.

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
