# Peer review of "Giant Bilateral Iliopsoas Abscesses, Secondary to Pott’s Disease: Challenging Diagnostic–Therapeutic Protocol Management (Modern and Innovative Open Approach Technique Through J.L. Petit Triangle)—A Case Report and Literature Review"

_diagnostics, 2025, doi:10.3390/diagnostics15192506_

Round 1
Reviewer 1 Report
Comments and Suggestions for Authors
This manuscript presents a rare and clinically important case of giant bilateral iliopsoas abscesses secondary to Pott’s disease, managed via an innovative bilateral JL Petit triangle approach. The main question addressed is whether this anatomical corridor is an effective and safe option for bilateral drainage in spinal tuberculosis. The topic is original and relevant, as bilateral abscesses are exceptionally uncommon, and this approach is rarely reported. Compared with existing literature, the authors contribute a novel surgical route, detailed interdisciplinary management, and long-term radiological and functional follow-up, while also drawing attention to diagnostic challenges such as a false-negative Quantiferon result.
The manuscript’s strengths include its clear clinical relevance, innovative anatomical approach, and comprehensive follow-up. Methodologically, the report would benefit from a clearer timeline of diagnostic steps, expanded operative details, and inclusion of objective functional outcome measures. The discussion should explore possible reasons for the negative Quantiferon result and compare the JL Petit triangle approach with other drainage techniques, highlighting advantages and limitations. Figures are helpful but could be improved by adding anatomical annotations or schematic diagrams; a comparative table summarizing prior bilateral cases would strengthen the literature review.
The conclusions are consistent with the presented data, though they should be tempered given the single-case design. References are largely appropriate. Language and formatting require minor corrections, including avoidance of abbreviations in the abstract, consistent font usage, and completion/clarification of the abbreviations list (e.g., BK, NIL, HIV).
Overall, this is a valuable and well-illustrated contribution that, with the suggested refinements, can provide meaningful insight into the field.
Author Response
Thank you very much for taking the time to review this manuscript. Please find the detailed responses below and the corresponding revisions and corrections in the re-submitted files.
Comments 1: The main question addressed is whether this anatomical corridor is an effective and safe option for bilateral drainage in spinal tuberculosis.
Response 1: This anatomical corridor is an effective and safe option for bilateral drainage in spinal tuberculosis, as evidenced by the good evolution, with the cure of the patient presented.
Comments 2: Methodologically, the report would benefit from a clearer timeline of diagnostic steps,
Response 2: The diagnostic stages were presented clearly and succinctly, chronologically, using the data extracted from the observation sheets, in dynamics, between February 2023 and September 2024.
Comment 3: Methodologically, the report would benefit from a ... expanded operative details,
Response 3: The operative details consisted of: incision of approximately 4 cm, above the iliac crest, in the middle 1/3, at the level of the lumbar triangle J.L. Petit. We penetrated behind the external oblique and in front of the abdominal transverse, we perforated the thoraco-lumbar fascia, then the psoas sheath at the lateral edge. We opened the abscess and evacuated the pus, 500 ml of purulent fluid were evacuated on the left side, respectively 300 ml on the right side; then we practiced excisional debridement, lavage with antiseptic substances, abdominal compression for complete drainage. The two cavities were drained, each of them, with two drainage tubes: a silicone one of 28 Fr Ch and a Foley catheter of 22 Fr Ch. And finally, we practiced lavage on drain tubes with antiseptic substances and then dressing.
Comments 4: ... and inclusion of objective functional outcome measures.
Response 4: Objective measures of functional outcomes are: general condition improved rapidly, purulent drainage decreased, later becoming seropurulent and then serous-fibrinous, within just a few days. Low back pain and bilateral lower limb pain improved, algo paresthesia and functional impotence of the left lower limb improved. Analgesic position in the bed plane in semi-sitting, with the left lower limb in semiflexion and the hip externally rotated, disappeared immediately postoperatively, mobilization and movements of the left lower limb quickly became possible, initially with a walking frame, but the neurosurgeon's recommendations were for absolute bed rest followed by lumbar orthosis until the discitis lesions healed - six months. Subsequently, muscle toning exercises with protection of the lumbar vertebral segment were recommended (lumbar orthosis during the day), until the end of the anti-tuberculosis treatment (one year), then medical gymnastics and kinesiotherapy for general toning and gentle joint gymnastics. Currently, the patient has limped gait and left hip pain, analgesic decrease in mobility.
Comments 5: The discussion should explore possible reasons for the negative Quantiferon result ...
Response 5: Diagnostic challenges, such as a false-negative Quantiferon tuberculosis gold test result, could be secondary to immunosuppression, laboratory technique (incorrect collection, transport or processing of the blood sample - delay or insufficient shaking of the containers), problems with the reagents used or calibration of the equipment, extrapulmonary tuberculosis (the immune response may be altered, in this situation, leading to a false negative result), human or administrative error (misinterpretation of results or confusion of evidence).
Comments 6: ... compare the JL Petit triangle approach with other drainage techniques, highlighting advantages and limitations.
Response 6: Thank you very much for focusing on this idea. It is very useful and welcome and we introduce the additions in the text, highlighting the advantages and limitations of the different drainage techniques: Abscess drainage can be performed surgically or minimally invasive, guided percutaneously by imaging (ultrasound, CT) or endoscopically (laparoscopic)
- Image-guided drainage can also be done under local anesthesia or sedation and has the following advantages: lower anesthetic-surgical risk than open surgery. It would be indicated in selected cases, in patients with immediate major vital risk (e.g. sepsis), until a curative intervention can be performed: dragged patients, in serious condition, with maximum anesthetic-surgical risk or in selected cases - well-defined, unilocular, located in an accessible position, with fluid collection and microbiological investigation of it (cultures with antibiogram). It requires prior imaging investigation: ultrasound or CT with precise determination of the location of the abscess, dimensions, appearance. Abscesses can drain by puncture with a needle or catheter. It can be left in place and washing - drainage can be performed if necessary. Advantages: low costs, lower risks, simplicity. If the abscess is old, with dense multiloculate content, or if the abscess is deep or in the vicinity of vital structures (vessels, nerves), anatomical access may be difficult and limited, and this type of drainage may be inadequate and insufficient.
- Laparoscopic drainage is performed under general anesthesia by orotracheal intubation and mechanical ventilation – it is less used than image-guided drainage in selected cases, large multilocular abscesses and which, more frequently, associate other intra-abdominal pathologies: appendicitis, diverticulitis, pelvic inflammatory disease, peritoneal tuberculosis. Indicated in more complex cases, when an intraperitoneal pathology is associated or when guided drainage has not given results.
- Classical drainage is indicated when the previous ones have failed, or are not possible, in large, multilocular abscesses, associated with necrosis or extensive lesions, severe complications such as peritonitis, sepsis, intraperitoneal diseases that cannot be addressed laparoscopically, vertebral bone tuberculosis, retroperitoneal tumors.
- In turn, abdominal drainage can be open (transperitoneal or extraperitoneal). Open surgical drainage can be performed by a retroperitoneal approach (through the iliac fossa), lumbar in the case of abscesses of renal etiology or intraperitoneal - simultaneously with the treatment of a primary gastrointestinal cause (appendicitis, diverticulitis).
- The surgical technique preferred and promoted by us is based on lumbar surgical drainage, through the J.L. Petit triangle. This approach provides direct access to the iliopsoas muscle and allows for effective drainage of the abscess, reducing the risk of complications and recurrence. Drainage through Petit triangle can be maintained for a longer period of time without the risk of major bleeding or retroperitoneal superinfection. Lavage - easy drainage, with antiseptics or antituberculosis drugs (e.g., rifampicin lavage). Recurrent abscess or relapse (less frequent than in image-guided drainage) can be resolved more easily by re-catheterization of the superficial orifice with a drain tube.
Comments 7: Figures are helpful but could be improved by adding anatomical annotations or schematic diagrams;
Response 7: Agree. Annotations have been made at your recommendation, to all images, which will certainly help to make it easier to understand the information presented, which is why thank you very much! Figure 1 – arrows indicate the two collections. Figure 2 – a. incision above the iliac crest, in the middle 1/3, at the level of the J.L. Petit lumbar triangle, b. double drainage of the remaining cavity, c. final appearance, after fixation of the drainage tubes (silicone tube and Foley probe). Figure 3 – resolution of abscesses under drainage. Figure 5 – the arrow indicates the area of the proposed approach.
Comments 8: ... a comparative table summarizing prior bilateral cases would strengthen the literature review.
Response 8: From what we have studied in the literature, with the help of several search engines, through several international databases, we have not identified the presentation of a similar case of bilateral ilio-psoas abscess. We have completed the paper, in this regard, with a paragraph that mentions that we have not found a similar case of bilateral abscess in the literature.
Comments 9: The conclusions are consistent with the presented data, though they should be tempered given the single-case design.
Response 9: The conclusions are consistent with the case presented and punctually reveal the justification of the title - Challenging Diagnostic-Therapeutic Protocols Management (Modern and Innovative Open Approach Technique Through J.L. Petit Triangle). Although a unique case design is presented, the peculiarities, not few, of diagnostic-therapeutic management of a complex case are presented. In addition, the paper presents a review of the literature.
Comments 10: Language and formatting require minor corrections ...
Response 10: Agree. We revised the language to be as clear and accurate as possible. We also revised the formatting to have a good manuscript format for printing.
Comments 11: ... including avoidance of abbreviations in the abstract, consistent font usage, and completion/clarification of the abbreviations list (e.g., BK, NIL, HIV).
Response 11: We removed all the abbreviations from the text, including the summary.
Thank you very much, once again, for the suggested recommendations!
We have attached an additional bibliography for a better understanding and explanation of the text:
- Centers for Disease Control and Prevention (CDC). QuantiFERON-TB Gold Plus (QFT-Plus) Fact Sheet, 2017, www.cdc.gov/tb/publications/factsheets/testing/quantiferon.htm
- World Health Organization (WHO). Latent Tuberculosis Infection: Updated and Consolidated Guidelines for Programmatic Management, 2018, www.who.int/publications/i/item/9789241550239
- Pai, M., Denkinger, C. M., Kik, S. V., et al. Gamma Interferon Release Assays for Detection of Mycobacterium tuberculosis Infection. Clinical Microbiology Reviews, 2014, Volume 27, pp 3–20. DOI: 10.1128/CMR.00034-13
- Lewinsohn, D. M., Leonard, M. K., LoBue, P. A., et al. Official American Thoracic Society/Infectious Diseases Society of America/CDC Clinical Practice Guidelines: Diagnosis of Tuberculosis in Adults and Children. Clinical Infectious Diseases, 2017, Volume 64, pp. 1-33 DOI: 10.1093/cid/ciw694
- Diel, R., Goletti, D., Ferrara, G., et al. Interferon-γ Release Assays for the Diagnosis of Latent Mycobacterium tuberculosis Infection: A Systematic Review and Meta-Analysis. European Respiratory Journal, 2011, Volume 37, pp. 88–99. DOI: 10.1183/09031936.00115110
Reviewer 2 Report
Comments and Suggestions for Authors
I would like to congratulate you on presenting this rare clinical case and review. Your effort in documenting and sharing this work is commendable, and it represents a valuable contribution to the literature. I have the following minor revision suggestions, which, if addressed, will improve the clarity of your manuscript;
1. Introduction: Please make the introduction more concise.
-
Clearly highlight the novelty of this case — specifically, bilateral giant iliopsoas abscess secondary to Pott’s disease managed with open drainage via the J.L. Petit triangle.
-
Provide a clear rationale for selecting this particular surgical approach
2. Case report:
- I would like to comment on the positioning of the surgical wound drain tube described in your study. It is generally recommended to place the drain tube through a separate stab incision site, slightly away from the primary surgical incision, using a subcutaneous tunnel. This technique is widely encouraged as it may reduce the risk of surgical site infection and promote better wound healing.
- However, in your report, the drain tube appears to have been placed directly through the main skin incision. Could you please clarify the rationale behind this choice?
3. Discussion:
-
Expand on the specific reasons for choosing the J.L. Petit triangle approach over alternative techniques, considering the patient’s characteristics (bilateral large abscesses, obesity, prior neurological conditions).
-
Briefly compare your approach with outcomes reported for image-guided or laparoscopic drainage.
Thank you for your valuable contribution and for sharing your findings with the scientific community.
Comments on the Quality of English LanguagePlease revise the manuscript for English grammar, syntax, and conciseness. Break down long sentences into shorter, clearer statements to improve readability.
Author Response
Thank you very much for taking the time to review this manuscript. Please find the detailed responses below and the corresponding revisions and corrections in the re-submitted files.
Comments 1: Please make the introduction more concise.
Response 1: Thank you very much for your advice! We took your recommendations into account and made the introduction more concise.
Comments 2: Clearly highlight the novelty of this case ...
Response 2: The particularities of this case consist of the fact that the abscesses were gigantic and present bilaterally, which we have not encountered in the specialized literature, and the novelty of drainage through the Petit triangle, an innovative and modern technique, has also not been cited.
Comments 3: Provide a clear rationale for selecting this particular surgical approach.
Response 3: We preferred lumbar approach (through J.L Petit triangle) because it offers direct drainage of the abscess with proven efficacy, minimizing the risk of hemorrhagic or infectious complications and recurrences.
Comments 4: Could you please clarify the rationale behind this choice?
Response 4: I would like to comment on the positioning of the drain tube. As you have observed, it gets used to place the drain tube in a different place, separate from the incision. We preferred to drain the remaining abscess cavity through the postoperative wound for several reasons:
- The drainage of the abscess should be as wide and direct as possible to allow continuous, easy and efficient postoperative lavage and drainage
- The rationale for placing trans-incisional drain tubes was to maintain as wide a direct communication as possible with the remaining abscess cavity for as long as possible and to promote healing from deep to superficial.
- To maintain the widest, lowest and longest access possible
- The advantage, even, of transforming into a tunnel over time, with progressive scarring, which, if necessary, could even allow easier re-catheterization.
- The anatomical peculiarity of the region, small in size, which can be approached safely without major risks, practically corresponding to the incision made.
- An additional incision in a septic area increases the risk of further contamination of that region
Comments 5: Expand on the specific reasons for choosing the J.L. Petit triangle approach over alternative techniques, considering the patient’s characteristics (bilateral large abscesses, obesity, prior neurological conditions). Briefly compare your approach with outcomes reported for image-guided or laparoscopic drainage.
Response 5: Your request is very useful and welcome, and we will insert the additions into the text, highlighting the advantages and limitations of the different drainage techniques.
- Abscess drainage can be performed surgically or minimally invasively, guided percutaneously by imaging (ultrasound, CT) or endoscopically (laparoscopic).
- Image-guided drainage can also be done under local anesthesia or sedation and has the following advantages: lower anesthetic-surgical risk than open surgery. It would be indicated in selected cases, in patients with immediate major vital risk (e.g. sepsis), until a curative intervention can be performed: patients with multiple comorbidities, in serious condition, with maximum anesthetic-surgical risk or in selected cases - well-defined, unilocular, well-defined abscess, located in an accessible position, with fluid collection and microbiological investigation of it (cultures with antibiogram). Prior imaging investigation is required: ultrasound or CT with precise determination of the location of the abscess, dimensions, appearance. Abscesses can be drained by needle or catheter puncture. It can be left in place and washing - drainage can be performed if necessary. Advantages: low costs, lower risks, simplicity. If the abscess is old, with dense multiloculate content, or if the abscess is deep or in the vicinity of vital structures (vessels, nerves), image-guided access may be difficult and limited, and this type of drainage may be inadequate and insufficient.
- Laparoscopic drainage is performed under general anesthesia by orotracheal intubation and mechanical ventilation – it is less used than image-guided drainage in selected cases, large multilocular abscesses and which, more frequently, associate other intra-abdominal pathologies: appendicitis, diverticulitis, pelvic inflammatory disease, peritoneal tuberculosis. Indicated in more complex cases, when an intraperitoneal pathology is associated or when guided drainage has not given results.
- Classical drainage is indicated when the previous ones have failed, or are not possible, in large, multilocular abscesses, associated with necrosis or extensive lesions, severe complications such as peritonitis, sepsis, intraperitoneal diseases that cannot be addressed laparoscopically, vertebral bone tuberculosis, retroperitoneal tumors. In turn, abdominal drainage can be open (transperitoneal or extraperitoneal). Open surgical drainage can be carried out by a retroperitoneal approach (through the iliac fossa), lumbar in the case of abscesses of renal or intraperitoneal etiology - simultaneously with the treatment of a primary gastrointestinal cause (e.g. appendicitis, diverticulitis).
- The surgical technique preferred and promoted by us is based on lumbar surgical drainage, through the J.L. Petit triangle. This approach provides direct access to the iliopsoas muscle and allows for effective drainage of the abscess, reducing the risk of complications and recurrence. Drainage through Petit triangle can be maintained for a longer period of time without the risk of major bleeding or retroperitoneal superinfection. Lavage - drainage are easy, with antiseptics or antituberculosis drugs (e.g., rifampicin lavage). Recurrent abscess or relapse (less frequent than in image-guided drainage) can be resolved more easily by re-catheterization of the superficial orifice with a drain tube.
Comments 6: Please revise the manuscript for English grammar, syntax, and conciseness. Break down long sentences into shorter, clearer statements to improve readability.
Response 6: Thank you very much for your advice! We have revised the manuscript according to your recommendations.